# Parameter Identification of Cutting Forces in Crankshaft Grinding Using Artificial Neural Networks

**DOI:** 10.3390/ma13235357

**Published:** 2020-11-26

**Authors:** Ivan Pavlenko, Milan Saga, Ivan Kuric, Alexey Kotliar, Yevheniia Basova, Justyna Trojanowska, Vitalii Ivanov

**Affiliations:** 1Department of General Mechanics and Machine Dynamics, Faculty of Technical Systems and Energy Efficient Technologies, Sumy State University, 2, Rymskogo-Korsakova St., 40007 Sumy, Ukraine; i.pavlenko@omdm.sumdu.edu.ua; 2Department of Applied Mechanics, Faculty of Mechanical Engineering, University of Zilina, 8215/1, Univerzitna St., 010 26 Zilina, Slovakia; milan.saga@fstroj.uniza.sk; 3Department of Automation and Production Systems, Faculty of Mechanical Engineering, University of Zilina, 8215/1, Univerzitna St., 010 26 Zilina, Slovakia; ivan.kuric@fstroj.uniza.sk; 4Department of Mechanical Engineering Technology and Metal-Cutting Machines, Institute of Education and Science in Mechanical Engineering and Transport, National Technical University “Kharkiv Polytechnic Institute”, 2, Kyrpychova St., 61002 Kharkiv, Ukraine; alexey_kotliar@ukr.net (A.K.); e.v.basova.khpi@gmail.com (Y.B.); 5Department of Production Engineering, Faculty of Mechanical Engineering, Poznan University of Technology, 5, M. Sklodowskej-Curie Sq., 60-965 Poznan, Poland; justyna.trojanowska@put.poznan.pl; 6Department of Manufacturing Engineering, Machines and Tools, Faculty of Technical Systems and Energy Efficient Technologies, Sumy State University, 2, Rymskogo-Korsakova St., 40007 Sumy, Ukraine

**Keywords:** technological process, intensification, grinding parameters, ANN model, regression approach

## Abstract

The intensifying of the manufacturing process and increasing the efficiency of production planning of precise and non-rigid parts, mainly crankshafts, are the first-priority task in modern manufacturing. The use of various methods for controlling the cutting force under cylindrical infeed grinding and studying its impact on crankpin machining quality and accuracy can improve machining efficiency. The paper deals with developing a comprehensive scientific and methodological approach for determining the experimental dependence parameters’ quantitative values for cutting-force calculation in cylindrical infeed grinding. The main stages of creating a method for conducting a virtual experiment to determine the cutting force depending on the array of defining parameters obtained from experimental studies are outlined. It will make it possible to get recommendations for the formation of a valid route for crankpin machining. The research’s scientific novelty lies in the developed scientific and methodological approach for determining the cutting force, based on the integrated application of an artificial neural network (ANN) and multi-parametric quasi-linear regression analysis. In particular, on production conditions, the proposed method allows the rapid and accurate assessment of the technological parameters’ influence on the power characteristics for the cutting process. A numerical experiment was conducted to study the cutting force and evaluate its value’s primary indicators based on the proposed method. The study’s practical value lies in studying how to improve the grinding performance of the main bearing and connecting rod journals by intensifying cutting modes and optimizing the structure of machining cycles.

## 1. Introduction

Among the priority tasks for the modern competitive machine-building industry, the active search for optimal technological solutions in intensifying technological processes is highlighted in [1,2]. Priority is given to improving the efficiency of manufacturing critical and expensive parts, such as the crankshaft of an internal combustion engine. Such parts are subject to high requirements in terms of roughness, dimensional accuracy, the accuracy of the shape of the main bearing and connecting rod journals, and their spatial position. These requirements are fulfilled by machining these parts using universal and particular grinding machines using special-purpose tooling. At the same time, it is necessary to note the sharply increased requirements for technical and economic indicators of grinding operations, especially in automated and robotic production. Therefore, under multiproduct manufacturing, it is necessary to consider the criteria of quality [3,4] and energy efficiency [5,6], equipment [7,8] and tooling [9,10] capabilities, processing modes [11,12], the design and technological parameters of the parts [13,14,15], the properties of materials and coatings [16,17,18,19,20].

The performance and cost of cylindrical infeed grinding operations for crankpins are determined mainly by the grinding cycle’s selected parameters and the method of managing this cycle. Please note that when grinding crankpins, the optimal machining cycle can be determined by the minimum machining time, which, in its turn, is determined by the speed of the cross-feed and grinding allowance. Increasing cross-traverse leads to an increase in the cutting-force components, which increases the elastic strain of the technological system elements. The elastic strain of the technological system elements and the crankshaft itself, which has variable rigidity depending on the rotation angle, leads to unacceptable processing errors. Thus, the cutting force is a limiting factor in improving crankpins’ grinding performance, which is why the development of methods for its rapid and accurate calculation is a high-priority task.

From the analysis of the existing systems for controlling the grinding cycle of complex parts, it is necessary to highlight the critical need to find fast, reliable, and cost-effective ways to adjust the machine control algorithm. It will have a definitive impact on the performance and quality of crankpin machining.

One of the promising ways to solve this problem is to use artificial intelligence tools to determine a mathematical model’s parameters for calculating cutting forces during grinding. In this case, it will also be sufficient to use the mathematical apparatus to obtain the summands of target functions, such as multi-parameter quasi-linear regression analysis.

## 2. Literature Review

Analysis of many scientists’ research results has shown that much attention is paid to the intensification of cylindrical infeed grinding processes and control of cutting forces in this machining method. In particular, S. da Silva et al. [21] noted that despite the existence of several models for effective grinding cycle design, they are not frequently adopted in production lines due to the lack of knowledge, difficulties in determining some input parameters, and discrepancies observed in the obtained results during the production of a batch of parts. That is why this difficult and time-consuming process requires careful preparation for implementing the machining of out-of-repair machine parts even at the stage of pre-production engineering.

It is a well-known fact that high grinding performance and ensuring the accuracy and quality of machining depend on a stable thermal regime in the cutting zone by efficiently removing the released heat. Concerning that, M. Stepanov et al. [22] have premised that the most instability is characteristic for the heat entering the machine tool from the coolant system. In this work, the potentialities of decreasing the influence of heat fluxes on the grinding machine’s accuracy by improving coolant tanks’ cooling ability were proposed. In particular, A. Patel et al. [23] consider that the speed ratio parameter significantly impacts workpiece surface roughness, workpiece texture, and power consumption. The work’s main achievement lies in the experimental determination that, compared to non-integer speed ratios, integer speed ratios yielded reduced surface roughness for the part and visible surface textures for the piece.

P. Lezanski [24] described the use of an artificial neural network (ANN) model to predict the wear propagation process of the grinding wheel and to estimate the remaining useful life of the wheel when the extrapolated data reaches a predefined final failure value. It allows consideration of the issue of developing an effective multilayer perceptron model and using it in the prediction of the remaining useful life for the grinding wheel is discussed. This study can be supplemented by M. Shapovalova et al. [25], where the architecture of a convolution neural network for image analysis of steel microstructure has been offered.

D. Lipinski et al. [26] presented the methodology of optimizing the sequential grinding process by applying fuzzy logic to define the machining process’s objectives and constraints. A. Boaron and W. Weingaertner [27] decided to intensify cylindrical infeed grinding due to acoustic emission-based quick test-method for in-process determination of the topographic characteristics of a fused grinding wheel. The authors argue that the presented method permitted obtaining the grinding wheel topography information at usual cutting speeds. B. Botcha [28] identified the physical relationship connecting the measured vibration signal with the surface characteristics based on the grinding process intensification. The model integrates the random distribution of the abrasive particles forming the wheel topography, the interactions between an elementary abrasive particle and the workpiece surface, and the regenerative relationship between the machine structure’s vibrations and the cutting forces has been developed.

Two approaches that minimize the grinding process’s safety margin, thus optimizing the process’s economic efficiency, have been introduced by M. Steffan et al. [29]. Both control concepts use the feed rate override of the machining operation as a regulating variable to eliminate the edge zone’s thermal damage. One result of this work has been the drafting of a control concept for grinding of noncircular workpieces, which revealed a potential for significant efficiency enhancement.

Ensuring crankshaft quality by abrasive machining methods was considered in the works [30,31,32]. Notably, F. Bordin et al. [30] proposed intensifying the machining process by varying the morphological characteristics of grinding wheels analyzed via X-ray tomography. The grinding force was monitored, and its components were determined. In the article [31], the number of studied process characteristics was expanded. Thus, studies of a dependence of the total cutting force for grinding wheels with a different grit on the material’s ultimate strength, the main bearing journal’s width, and infeed speed were made. In the research [32], recommendations for determining the optimal parameters for the cylindrical infeed grinding cycle of the crankpins from productivity and accuracy were developed.

Significant experience has been gained in designing diamond wheels [33,34]. Furthermore, a novel approach for investigating the influence of diamond grain wear during grinding on the grain is suggested. The role of such factors as grain grade and wheel bond, relative orientation, and degree of diamond grain wear can be evaluated during the development, production, and operation of diamond composite materials [35]. The received theoretical regulations have a fundamental nature and can be used in industry for machine design [36].

Maier, M. et al. [37] implemented data-driven optimization methods for studying the grinding process. In particular, a self-optimization algorithm of a grinding machine was developed for decreasing production costs. P. Hernández-Becerro et al. [38] studied the thermo-mechanical response of a five-axis precision machine tool to the temperature change. As a result, the implemented reduced-order model based on the Monte Carlo simulation allowed the carrying out of parameter identification of the proposed model.

The scientific approach for solving a problem of vibration reliability of machines based on artificial neural networks is developed [39]. The proposed methodology integrates analytical dependencies, novel techniques of numerical simulations, and artificial neural networks [40]. The scientific novelty of the proposed method based on parameter identification lies in the inconsequent implementation of the numerical analysis approach, mathematical modeling of processes using the quasi-linear regression procedure, and artificial intelligence systems [41].

However, all these works do not provide a single approach for the possibility of intensifying the machining of the part’s surface as early as at the planning stage of the machining process, which in turn complicates the issue of cost-effectiveness of the crankshaft production process. Moreover, there is no single approach to evaluating process parameters based on a reliable mathematical model for determining the cutting force.

## 3. Materials and Methods

### 3.1. General Formulation of the Problem

Experimental studies of the process of cylindrical infeed grinding, performed by M. Stepanov and L. Khodakov at the machine-tool laboratory of the machine-tool building plant named after S.V. Kosior (Kharkiv, Ukraine), have revealed the following empirical dependence for determining the circumferential component *P_z_* of the cutting force [31]:(1)$$P_{z}=\alpha \frac{\sigma_{t}^{\beta_{1}}\cdot H^{\beta_{2}}\cdot V_{p}^{\beta_{3}}}{Z^{\beta_{4}}\cdot S^{\beta_{5}}\cdot S_{pr}^{\beta_{6}}\cdot t_{pr}^{\beta_{7}}}\cdot B^{\beta_{8}},$$
where *σ_t_*—ultimate strength of workpiece material at a high temperature (about 600 °C), kgf/mm^2^; *H*—sonic index of the grinding wheel; *Z*—grinding wheel grit, μm; *V_p_*—infeed speed, mm/min; *S*—the peripheral speed of workpiece rotation, m/min; *S_pr_*—the longitudinal speed of grinding wheel dressing, mm/min; *t_pr_*—dressing depth, mm; *B*—grinding width, mm.

Formula (1) contains one unknown coefficient *α* and *m* = 8 of unknown indices of power *β_k_* (*k* = 1, 2,…, *m*), defined for specific technological conditions of production. However, there is no general method for determining these parameters.

Given the above, the purpose of this work is to create a comprehensive scientific and methodological approach for determining the quantitative values of the dependence parameters (1). This goal is achieved by a consistent implementation of the following stages of scientific research:
creating a method for conducting a virtual experiment to determine the cutting force depending on an array of *m* parameters of formula (1):
creating a table of input data for the variation range of parameters that affect the cutting force;generating an array of experimental data as a sample of a sufficiently large number of n random arrays of input parameters:
creating a subroutine for determining the total number of each input parameter;generating a set of n combinations of values of m input parameters with a predetermined relative error δ;calculation of n values of cutting forces as a result of a virtual experiment;determining the maximum values of each input parameter and the cutting force $P_{z}^{max}$;
using artificial intelligence tools to identify parameters of a mathematical model based on experimental data:
normalization of input and output parameters;creating an artificial neural network and configuring its parameters;training an artificial neural network based on an array of normalized experimental data;determining the accuracy of estimating the cutting-force value for an arbitrary set of input parameters;creating a reliable generalized mathematical model for estimating (*m* + 1) parameters that determine the cutting force using multi-parameter quasi-linear regression analysis:
creating a matrix relation for determining the cutting force based on experimental data;formation of the stiffness matrix and the column vector of external influence:formation of column sub-vectors of external influence and local stiffness parameters;formation of the stiffness submatrix;globalization of submatrix and column sub-vectors to a common stiffness matrix;using the inverse matrix method to evaluate (*m* + 1) unknown parameters;forming a ratio for calculating the cutting force based on a specific coefficient and indices of power, and comparing the obtained dependence with the empirical formula (1);determination of relative errors in determining unknown coefficients of the regression model.

Notably, the last item’s implementation into the main sequence forms a major advantage of the proposed method for estimating the mathematical model parameters compared to the typical ANN procedure.

### 3.2. Virtual Experiment on the Cutting Force

Experimental studies of the cutting force were carried out at the Research Laboratory of the “Kharkiv Machine-Tool Building Plant named after S. V. Kosior” by carrying out multifactorial experiments on a CNC (computer numerical control) cylindrical grinding machine. As a result of experiments for various steels’ grades, grinding wheels have made of electrocorundum of various grain sizes were used. Additionally, the cutting parameters were varied in a range of their possible values.

The range of values and the increment of these eight parameters are shown in Table 1. The range of values and the increment of these parameters’ changes are determined by the cutting modes, manufacturing engineering technology, and technological feasibility.

Specially created subroutines for determining each input parameter’s total number and generating a sample from a given set of experimental data from Table 1 allows the generation of a set of combinations of input parameter values. At that, the following symbols are introduced in order to unify further calculations: *x*_1_ = *σ_t_*, *x*_2_ = *H*, *x*_3_ = *V_p_*, *x*_4_ = *Z*, *x*_5_ = *S*, *x*_6_ = *S_pr_*, *x*_7_ = *t_pr_*, *x*_8_ = *B*.

The total number of experiments is *n* = 100, and the relative error in determining each of *m* = 8 parameters does not exceed 3%. The virtual modeling resulted in an array of *n* × (*m* + 1) parameters with the following structure:(2)$$\left[M\right]=\left[\begin{array}{c} x_{1}^{<1>}\;\;x_{2}^{<1>}\;\;\ldots \;\;x_{m}^{<1>}\;\;|\;\;y^{<1>}\\ x_{1}^{<2>}\;\;x_{2}^{<2>}\;\;\ldots \;\;x_{m}^{<2>}\;\;|\;\;y^{<2>}\\ \ldots \\ x_{1}^{<i>}\ldots \;\;x_{k}^{<i>}\ldots \;x_{m}^{<i>}\;|\;\;y^{<i>}\\ \ldots \\ x_{1}^{<n>}\;\;x_{2}^{<n>}\;\;\ldots \;\;x_{m}^{<n>}\;|\;\;y^{<n>} \end{array}\right],$$
where $\;x_{k}^{<i>}$—*k*-th parameter of the *i*-th experiment (*k* = 1, 2, …, *m*; *i* = 1, 2, …, *n*); $y^{<i>}$—force value *P_z_*, determined as a result of conducting the *i*-th experiment.

## 4. Results

### 4.1. Application of Artificial Intelligence Systems

The procedure of normalization of all elements of the experimental data array is performed beforehand to use artificial intelligence tools, in particular, artificial neural networks:(3)$$\;{\hat{x}}_{k}^{<i>}=\frac{\;x_{k}^{<i>}}{\;x_{k}^{max}};\;{\hat{y}}^{<i>}=\frac{y^{<i>}}{y^{max}},$$
where $y^{max}$ = $P_{z}^{max}$ = 153 H—maximum value of the cutting force. The maximum values $\;x_{k}^{max}$ of each input parameter *x_k_* are shown in Table 1.

The result is an array of *n* × (*m* + 1) normalized parameters with the following structure:(4)$$\left[\hat{M}\right]=\left[\begin{array}{c} {\hat{x}}_{1}^{<1>}\;\;{\hat{x}}_{2}^{<1>}\;\;\ldots \;\;{\hat{x}}_{m}^{<1>}\;\;|\;\;{\hat{y}}^{<1>}\\ {\hat{x}}_{1}^{<2>}\;\;{\hat{x}}_{2}^{<2>}\;\;\ldots \;\;{\hat{x}}_{m}^{<2>}\;\;|\;\;{\hat{y}}^{<2>}\\ \ldots \\ {\hat{x}}_{1}^{<i>}\ldots \;\;{\hat{x}}_{k}^{<i>}\ldots \;{\hat{x}}_{m}^{<i>}\;|\;\;{\hat{y}}^{<i>}\\ \ldots \\ {\hat{x}}_{1}^{<n>}\;\;{\hat{x}}_{2}^{<n>}\;\;\ldots \;\;{\hat{x}}_{m}^{<n>}\;|\;\;{\hat{y}}^{<n>} \end{array}\right],$$

The corresponding array of normalized values was created using a set of experimental data (Table 2).

Notably, the heat effect on cutting force was not considered due to the coolant’s intensive impact.

The experimental data is used to train an artificial neural network created using the “Visual Gene Developer”^®^ software. The architecture of the ANN is shown in Figure 1 a.

In particular, in this case, 2 hidden neural layers are selected, the first of which contains 5 neurons, and the second—3 neurons.

To train an artificial neural network with an array of normalized experimental data, the following settings are selected: learning rate—0.01; momentum coefficient—0.1; transfer function—hyperbolic tangent; the maximum number of training cycles—1 × 10^6^; target error—1 × 10^−5^; initialization method of the threshold—random; initialization method of weighting factor—random; analysis update interval—500 cycles.

As a result of training, the following convergence parameters are obtained: the sum of error—1.6 × 10^−3^; the average error per output per dataset—1.6 × 10^−5^. Thus, the training accuracy of an artificial neural network is relatively high.

It should be noted that the following characteristics confirm the high accuracy of estimating the value of the cutting force for an arbitrary set of input parameters using artificial intelligence: regression coefficient—0.9994; slope—0.9993; y-intercept—0.0007. It confirms the validity of the generated regression model. Visualization of the regression procedure’s convergence process resulting from training an artificial neural network is shown in Figure 1b.

### 4.2. Multi-Parameter Quasi-Linear Regression Procedure

Despite the high accuracy of using artificial intelligence tools to determine the cutting force’s dependence on the array of input parameters, it does not directly establish the corresponding functional dependence. However, this problem can be solved by using a multi-parameter quasi-linear regression procedure.

Thus, according to the generally accepted approach in mechanical engineering, justified in this case by formula (1), any estimated value of y (for example, the cutting force) can be determined from an array of input parameters *x_k_* using the following dependence:(5)$$y=\alpha \overset{m}{\underset{k=1}{\prod }}x_{k}^{\beta_{k}},$$
which is a generalization of formula (1).

Since this relation is nonlinear, evaluation of its parameters α, $\beta_{k}$ is performed by logarithm; as a result, formula (5) allows the obtaining of the corresponding quasi-linear model:(6)$$\bar{y}=\bar{\alpha }+\overset{m}{\underset{k=1}{\sum }}\beta_{k}{\bar{x}}_{k},$$
where the following symbols are introduced:(7)$$\bar{y}=lny;\;\bar{\alpha }=ln\alpha ;\;{\bar{x}}_{k}=lnx_{k}.$$

The parameters $\bar{\alpha }$, ${\bar{x}}_{k}$ are evaluated by minimizing the target function of the total square deviation [42]:(8)$$R(\bar{\alpha },\{\bar{\beta }\})=\overset{n}{\underset{i=1}{\sum }}{\left(\bar{\alpha }+\overset{m}{\underset{k=1}{\sum }}{\bar{x}}_{k}^{\left\langle i\right\rangle }{\bar{\beta }}_{k}\right)}^{2}\to min,$$
where $\left\{\bar{\beta }\right\}$—column vector of estimated indices of power $\beta_{k}$; ${\bar{x}}_{k}^{<i>}$—the logarithm of the value of the *k*-th input parameter within the *i*-th experiment.

The condition of a minimum of the target function *R* is the system of equations [43]
(9)$$\left\{\begin{array}{c} \frac{\partial R}{\partial \bar{\alpha }}=0;\\ \frac{\partial R}{\partial \left\{\bar{\beta }\right\}}=0, \end{array}\right.$$
which, considering formula (8), takes the following expanded form:(10)$$\left\{\begin{array}{c} \overset{n}{\underset{i=1}{\sum }}\left(\bar{\alpha }+\overset{m}{\underset{k=1}{\sum }}{\bar{x}}_{k}^{\left\langle i\right\rangle }{\bar{\beta }}_{k}\right)=0;\\ \overset{n}{\underset{i=1}{\sum }}\left(\bar{\alpha }+\overset{m}{\underset{k=1}{\sum }}{\bar{x}}_{k}^{\left\langle i\right\rangle }{\bar{\beta }}_{k}\right){\bar{x}}_{j}^{\left\langle i\right\rangle }=0, \end{array}\right.$$
where *j*—the number of input parameter (*j* = 1, 2, …, *m*).

The last system of (*m* + 1) linear algebraic equations concerning parameters $\bar{\alpha }$, ${\bar{\beta }}_{k}$ (*k* = 1, 2,…, *m*) can be written in the following matrix form:(11)$$\left[C\right]\left\{X\right\}=\left\{Y\right\},$$
where [*C*], {*X*}, {*F*} are the stiffness matrix, the column vector of external influence, and the column vector of the required parameters, respectively $\bar{\alpha }$, ${\bar{\beta }}_{k}$:(12)$$\left[C\right]=\left[\begin{array}{ccc} n & | & {\left\{S\right\}}^{T}\\ -- & | & --\\ \left\{S\right\} & | & \left[D\right] \end{array}\right];\left\{X\right\}=\left\{\begin{array}{l} \bar{\alpha }\\ \left\{B\right\} \end{array}\right\};\left\{F\right\}=\left\{\begin{array}{l} Y_{0}\\ \left\{Y\right\} \end{array}\right\},$$
which in their structure contain the element *Y*_0_, as well as the submatrix [*D*] and the column sub-vectors {*S*} and {*Y*}, whose elements are defined by the following formulas:(13)$$Y_{0}=\overset{n}{\underset{i=1}{\sum }}{\bar{y}}^{\left\langle i\right\rangle };\;Y_{k}=\overset{n}{\underset{i=1}{\sum }}{\bar{y}}^{\left\langle i\right\rangle }{\bar{x}}_{k}^{\left\langle i\right\rangle };\;S_{k}=\overset{n}{\underset{i=1}{\sum }}{\bar{x}}_{k}^{\left\langle i\right\rangle };\;D_{j,k}=\overset{n}{\underset{i=1}{\sum }}{\bar{x}}_{j}^{\left\langle i\right\rangle }{\bar{x}}_{k}^{\left\langle i\right\rangle }.$$

Using the inverse matrix method allows the setting of the column vectors of the values of the required parameters [44]:(14)$$\left\{X\right\}={\left[C\right]}^{-1}\left\{Y\right\}.$$

In this case, the total error of estimating the cutting force is determined by the formula:(15)$$\delta_{Pz}=\frac{1}{P_{z}}\sqrt{{\left(\alpha \frac{\partial y}{\partial \alpha }\delta_{\alpha }\right)}^{2}+\overset{m}{\underset{k=1}{\sum }}{\left(\beta_{k}\frac{\partial y}{\partial \beta_{k}}\delta_{\beta k}\right)}^{2}},$$
which considers expression (5) and, after identical transformations, takes the following form:(16)$$\delta_{Pz}=\sqrt{\delta_{\alpha }^{2}+\overset{m}{\underset{k=1}{\sum }}{\left(\beta_{k}lnx_{k}\right)}^{2}\delta_{\beta k}^{2}}.$$

For the array of experimental data shown in Table 2, the values of the required parameters determined by the implementation of a multi-parameter quasi-linear regression procedure are obtained. Therefore, for the total number of experiments *n* = 3 × 10^4^, the calculation results, including the estimation procedure’s errors, are summarized in Table 3.

Negative values of the indices of power *β*_4_, *β*_5_, *β*_6_, *β*_6_ confirm the inversely proportional dependence of the cutting force *P_z_* on the parameters Z, *S*, *S_pr_*, *t_pr_* in formula (1).

A relatively small measurement error confirms the reliability of the results obtained. Thus, the relative error in estimating the cutting force of coefficient *α* is 2.7%. The relative error in estimating the indices of power *β_k_* is in the range of 0.8–1.5%. As a result, the total relative error in determining the cutting force does not exceed 5.0%.

The input data have been varied in a range of 3% according to the uniform probability distribution law to study the effect of varying parameters on the evaluated data disturbances. As a result, the evaluated parameters’ average disturbances are in a range of 0.6–1.9%. Notably, the grinding width *B* and the infeed speed *V_p_* significantly impact the evaluated data (on average, 1.9% and 1.6%, respectively).

Finally, in production conditions, the proposed method allows the rapid and accurate assessment of the influence of the workpiece’s parameters, tool, and the machine on the power characteristics’ values for the cutting process. Its implementation requires experimental and statistical values of all necessary parameters and software for electronic data processing and the calculation process’s automation.

## 5. Conclusions

Thus, in the presented research, a method was developed to calculate and predict indicators that affect the cutting-force components during grinding to improve crankshaft machining efficiency. Simultaneously, accurate and fast calculation of cutting-force values depending on the production conditions makes it possible to adaptively control the crankpin grinding cycle’s parameters and ensure the machining intensification.

The developed approach is based on creating a comprehensive scientific and methodological approach for determining the quantitative values of parameters that determine the cutting force. As a result, a method for conducting a virtual experiment to determine the cutting force depending on the array of defining parameters was created. Moreover, artificial intelligence tools and multi-parameter quasi-linear regression analysis made it possible to identify these parameters. A special feature of the proposed approach lies in the possibility of considering the accuracy of determining unknown coefficients of the regression model and controlling the total error in estimating the cutting force.

The reliability of the proposed approach of using artificial neural networks is confirmed by the high value of Pearson’s product-moment correlation coefficient, which is equal to 0.9994. In general, the reliability of the proposed mathematical model for determining the cutting force using regression analysis is confirmed by the fact that for a given set of experimental data, the relative error in estimating parameters that affect the cutting force does not exceed 1.5%. As a result, the total relative error in determining the cutting force does not exceed 5.0%.

Finally, the developed scientific and methodological approach can be applied in a wide range of further research fields. In particular, for correctly formed sets of experimental data, the suggested algorithm of complex application of artificial intelligence tools and quasi-linear regression analysis can be used not only for evaluating the components of cutting forces of turning, milling, drilling, etc. but also for assessing the efficiency of technological processes in machines and apparatus.

## Figures and Tables

**Figure 1 materials-13-05357-f001:**
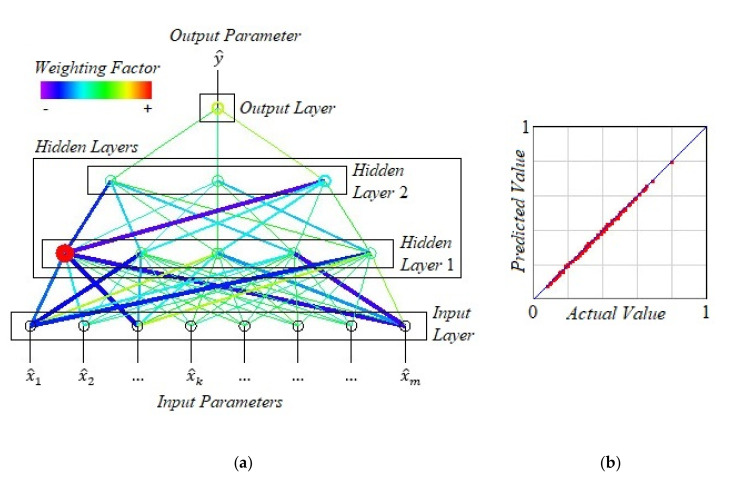
Artificial neural network architecture (**a**) and convergence of the regression procedure (**b**).

**Table 1 materials-13-05357-t001:** Range of values and increments of changes of grinding parameters.

Parameter	*σ_t_*	*H*	*V_p_*	*Z*	*S*	*S_pr_*	*t_pr_*	*B*
Measurement units	kgf/mm^2^	–	mm/min	μm	m/min	mm/min	mm	mm
Minimum value	20	1.38	0.10	16	20	60	0.01	20
Maximum value	120	1.60	0.15	40	100	300	0.03	120
Parameter change increment	10	0.02	0.01	8	5	10	0.01	2

**Table 2 materials-13-05357-t002:** Part of the experimental data array.

σ_t_	H	V_p_	Z	S	S_pr_	t_pr_	B	P_z_
kgf/mm^2^	–	mm/min	μm	m/min	mm/min	mm	mm	H
20	1.40	0.11	25	65	190	0.01	110	45
60	1.46	0.14	16	35	260	0.01	30	24
40	1.54	0.13	32	100	300	0.02	100	57
80	1.56	0.11	40	95	60	0.02	120	83
20	1.52	0.13	16	35	60	0.03	30	17
**Unitless Parameters**								
${\hat{x}}_{1}$	${\hat{x}}_{1}$	${\hat{x}}_{1}$	${\hat{x}}_{1}$	${\hat{x}}_{1}$	${\hat{x}}_{1}$	${\hat{x}}_{1}$	${\hat{x}}_{1}$	$\hat{y}$
0.167	0.875	0.733	0.625	0.650	0.633	0.333	0.917	0.294
0.500	0.913	0.933	0.400	0.350	0.867	0.333	0.250	0.157
0.333	0.963	0.867	0.800	1.000	1.000	0.667	0.833	0.373
0.667	0.975	0.733	1.000	0.950	0.200	0.667	1.000	0.542
0.167	0.950	0.867	0.400	0.350	0.200	1.000	0.250	0.111

**Table 3 materials-13-05357-t003:** Estimation results of formula parameters (5).

Parameter	*α*	*β* _1_	*β* _1_	*β* _1_	*β* _1_	*β* _1_	*β* _1_	*β* _1_	*β* _1_
Predicted value	2.315	0.337	0.256	0.932	−0.051	−0.072	−0.072	−0.025	0.985
Actual value	2.254	0.342	0.258	0.945	−0.051	−0.072	−0.072	−0.026	1.000
Relative error, %	2.7	1.4	0.9	1.4	0.8	1.2	1.5	2.0	1.5

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
