# Peer review of "Parameter Identification of Cutting Forces in Crankshaft Grinding Using Artificial Neural Networks"

_materials, 2020, doi:10.3390/ma13235357_

Round 1
Reviewer 1 Report
Dear Authors,
my comments are as follows:
- The title is not appropriate. The method of identification should be included, eg., ... using artificial neural networks.
- l. 22 should be "in" instead of "under"
- l. 51, 51: minimum machining time?
- The introduction is too long. Only papers related to the main topic should be included. `
- l. 151-182: what is the novelty of the proposed stages in comparison to a typical procedure with ANN?
- Table 2: How the experimental data were obtained? What were the experimental conditions, machine tools, tools, etc?
- Chapter 3.2: Is there any novelty in the presenteded theory?
- Conlcusions: what is the novelty of the developed method?
Author Response
Dear Reviewer,
Thank you very much for the time devoted to reviewing the manuscript.

Reviewer 2 Report
This paper proposes an optimization of grinding parameters, following a method based on learning with artificial neural network (ANN) and multiparametric quasilinear regression analysis to find the most appropriate values in a mechanistic model. The method is fully presented in simulation, and its transfer to a real case is not clarified. Several questions and improvements are suggested which should be answered to improve the quality of the paper.
1) The authors should discuss the uncertainty in their modeling and clarify the effect of unmodelled mismatch, noise and disturbanices on the robustness of the proposed ‘control’ method.
2) What is the effect of adding noise to the experimental data? How accurate are the prediction in this case? How accurate are the predictions in case of drift due to thermal/environmental/wearing effects?
3) The authors should perturb each of the parameters that they set in the generation of the experimental data and check how the ANN model performs, and provide the results to judge better the performance.
4) It looks like the proposed method provides an extremely good fit of the generated ‘experimental’ data, which looks like overfitting to the training data. Why not using a simpler method (e.g. nonlinear least squares minimisation) for the modeling?
5) What are the expected strengths and drawbacks of the method when used on a real system, and challenges when bringing the method to reality?
6) The authors completely omit the discussion of data-driven optimization methods for parameter set-up in grinding. A recent example which should be discussed is Maier, M., Rupenyan, A., Bobst, C. et al. Self-optimizing grinding machines using Gaussian process models and constrained Bayesian optimization. Int J Adv Manuf Technol 108, 539–552 (2020) as it includes in particular temperature monitoring near the grinding surface.
Author Response

(The authors gave the same response as above.)

Reviewer 3 Report
Artificial Neural Networks and Machine Learning have already become a pretty popular topic. It is very interesting to see how the authors apply this up-to-date method in the study of crankshaft grinding. Based on your methodology discussion, it can be applied to many research fields. I did not find big flaws in your article. I think, nonetheless, that the manuscript could be improved if the authors could address the comments and recommendations I listed in the following. 1.The novelty of this research should be highlighted in the Abstract. 2. Substitute a high-resolution figure in your figure 3. Give a citation to your equations. 4. Why you choose this model? Are you inspired by other people’s work?
Author Response

(The authors gave the same response as above.)

Round 2
Reviewer 1 Report
It can be accepted in the current form.
Author Response
The authors appreciate the reviewer for his/her contribution.
Reviewer 2 Report
The paper is improved after addressing the proposed feedback.
It is still not clear if varying the parameters with 3% has a significant effect or not.
on line 331: Notably, during the mathematical model development, all the input parameters are varied in a range of 3 % according to the uniform probability distribution law. However, these disturbances do not lead to the robustness of the proposed method.
Can the author rephrase this to make it clear?
Author Response
To clarify the effect of varying parameters on disturbances of the evaluated data, the corresponding sentences has been rephrased by the following one: “The input data have been varied in a range of 3 % according to the uniform probability distribution law to study the effect of varying parameters on the evaluated data disturbances. As a result, the evaluated parameters' average disturbances are in a range of 0.6–1.9 %. Notably, the grinding width B and the infeed speed Vp significantly impact the evaluated data (on average, 1.9 % and 1.6 %, respectively).”.
Additionally, a thorough spell-check has been carried out again.